# Blockchain-Based Electronic Voting: A Secure and Transparent Solution

Bruno Miguel Batista Pereira [1,*], José Manuel Torres [1,2], Pedro Miguel Sobral [1,2], Rui Silva Moreira [1,2], Christophe Pinto de Almeida Soares [1,2] and Ivo Pereira [1,3]

1    Intelligent Sensing and Ubiquitous Systems Group (ISUS), University Fernando Pessoa, Praça 9 de Abril, 4249-004 Porto, Portugal; csoares@ufp.edu.pt (C.P.d.A.S.); ivopereira@ufp.edu.pt (I.P.)
2    Artificial Intelligence and Computer Science Laboratory (LIACC), University of Porto, 4150-181 Porto, Portugal
3    Interdisciplinary Studies Research Center (ISRC), Instituto Superior de Engenharia do Porto (ISEP), Polytechnic of Porto, 4249-015 Porto, Portugal
*    Correspondence: 38054@ufp.edu.pt; Tel.: +351-225-071-300

**Abstract:** Since its appearance in 2008, blockchain technology has found multiple uses in fields such as banking, supply chain management, and healthcare. One of the most intriguing uses of blockchain is in voting systems, where the technology can overcome the security and transparency concerns that plague traditional voting systems. This paper provides a thorough examination of the implementation of a blockchain-based voting system. The proposed system employs cryptographic methods to protect voters' privacy and anonymity while ensuring the verifiability and integrity of election results. Digital signatures, homomorphic encryption (He), zero-knowledge proofs (ZKPs), and the Byzantine fault-tolerant consensus method underpin the system. A review of the literature on the use of blockchain technology for voting systems supports the analysis and the technical and logistical constraints connected with implementing the suggested system. The study suggests solutions to problems such as managing voter identification and authentication, ensuring accessibility for all voters, and dealing with network latency and scalability. The suggested blockchain-based voting system can provide a safe and transparent platform for casting and counting votes, ensuring election results' privacy, anonymity, and verifiability. The implementation of blockchain technology can overcome traditional voting systems' security and transparency shortcomings while also delivering a high level of integrity and traceability.

**Keywords:** blockchain; voting; genetic algorithm; electronic voting

---

## 1. Introduction

The adoption of a blockchain-based system for internal corporate use has emerged as a possible solution to a range of organizational difficulties. Conventional centralized systems frequently need a high level of confidence between participants and are prone to a variety of security and transparency challenges, including fraud, hacking, and data manipulation. Blockchain technology, on the other hand, provides a distributed, decentralized, and transparent platform for secure and efficient data sharing and tracking. According to Satoshi Nakamoto, the pseudonymous developer of Bitcoin, blockchain technology "*allows direct online transfers from one party to another without the need for a financial institution*" [1]. Since the publication of this white paper, blockchain technology has progressed beyond its initial usage for Bitcoin and has been researched for a variety of business systems.

This paper proposes a solution for the scheduling of courses in a university. It is a complex and tedious task that requires a lot of time and effort from both students and faculty staff. To address this issue, a system has been developed that generates and validates automatically different schedule options for the university. Further, the optimal solution may be elected by professors through the integration of a blockchain system.





The system is divided into two main parts: (i) the grouping of constraints that serve as an input for the generation of different possible schedules; and (ii) a blockchain voting system. The scheduling system consists of two types of restrictions: hard restrictions, which cannot be violated, and soft restrictions, which can be relaxed to some extent but are not mandatory. These constraints are incorporated into a genetic algorithm that generates a series of schedules, which are then integrated with the blockchain voting system.

The integration of blockchain technology into the scheduling system provides an additional layer of security and transparency. Using a decentralized network ensures the integrity of the schedule generated by the system, and any changes made to the schedule can be traced back to its source. This feature is particularly useful in situations where there is a need to maintain the confidentiality and privacy of data. The use of blockchain technology in the scheduling system also provides an immutable record of the voted schedules, which can be used for auditing purposes. Overall, integrating blockchain technology in the scheduling system provides a more secure and transparent solution for scheduling courses in a university.

Both parts are planned to work seamlessly together, constituting a unified system that addresses the challenges of both electronic voting (e-voting) and scheduling. The interconnection between the two systems is achieved through the use of shared data, which stores all the relevant data required for both systems. The voting system is planned to use the scheduling system to determine the optimal voting time based on the availability of voters and polling stations. On the other hand, the scheduling system is projected to use the voting system to ensure that no classes are scheduled during voting hours. This interconnection ensures that the two systems work together seamlessly, resulting in an efficient and effective system.

Several research studies have highlighted the importance of integrating e-voting and scheduling systems to improve efficiency and reduce costs. For example, the study "Genetic Algorithms Applied to University Exam Scheduling" by Jenkins et al. [2] used genetic algorithms to optimize the university timetable, resulting in an effective generation of high-quality schedules that met multiple objectives, such as minimizing the number of conflicts, maximizing the number of students who could take all their exams, and reducing the overall number of exam slots required. The results also indicate that the proposed approach in the paper can reduce the number of conflicts by up to 50% compared to previous approaches, and can improve the overall quality of exam schedules.

Similarly, in the study of Deris et al. entitled "A Constraint-Directed Genetic Algorithm (CD-GA) for University Timetable Planning" [3], the authors solved the university timetable planning problem using a CD-GA and evaluated it on benchmark datasets, and compared it with other timetable scheduling algorithms. The results show that the CD-GA can generate high-quality solutions in a reasonable time frame and is more efficient and effective than other algorithms.

The blockchain aspect will be the primary focus of this study, although the generation of timetables will also be considered as an important component. The blockchain implementation and optimizations in comparison to other projects are discussed. Observing how such a project performs in the real world can be instructive.

This paper is organized as follows. Section 2 includes a review of the literature and related work, identifying relevant scientific works and summarizing the market study, as well as a brief overview of similar projects. Section 3 describes the proposed system architecture, focusing on the scheduling system and the voting system. Section 4 discusses the use case, a real-world scenario in which the proposed system was implemented, and their results. Finally, Section 5 points out the conclusions and suggests some future work.

*Base of Theory*

Theoretical foundations are critical for understanding the motivation and possible impact of a project like this one. There has been an increase in interest in the use of genetic algorithms to solve different optimization issues, particularly scheduling challenges. Since

genetic algorithms can effectively explore the solution space and discover the best potential answers, they are particularly well-suited for optimization problems with many alternative solutions. Furthermore, the use of blockchain technology for decentralized decision-making and voting systems is gaining popularity as a way to ensure transparency, security, and immutability in a range of scenarios.

The integration of these two technologies into an academic scheduling system is a new method/approach to address a difficult challenge. The possible impact of this research includes academic scheduler optimization, which can lead to better student results and higher teacher satisfaction, as well as the study of the possibilities of blockchain technology in the context of decentralized decision-making systems.

## 2. State-of-the-Art Review and Related Works

Blockchain technology has grown in popularity in a variety of industries, including voting systems. The blockchain is a distributed ledger that enables the construction of tamper-resistant, transparent, and unchangeable records. The blockchain can establish a system that is very resistant to manipulation and fraud by employing cryptographic algorithms to safeguard transactions. Numerous researchers have proposed blockchain-based voting systems to improve election security and transparency. The potential for openness and security is one of the primary benefits of blockchain-based voting systems. According to Khan et al. [4], blockchain technology has the potential to improve voting security and transparency by enabling tamper-proof recordings of each vote.

Similarly, Jadhav et al. [5] contend that blockchain-based voting systems can eliminate traditional voting systems concerns such as hacking, tampering, and fraud.

The Avalanche blockchain protocol [6] is projected to provide fast and secure transactions for large-scale distributed computing systems. It introduces a consensus algorithm called Avalanche, which enables nodes in the network to quickly reach a consensus on the state of the ledger. The algorithm relies on a random sampling mechanism to select a small subset of nodes for the consensus process. The Avalanche protocol is planned to be highly scalable and can handle complex smart-contracts (SS) efficiently. The smart-contracts concepts will be further discussed and reviewed in Section 2.2.2.

In 2016, Kartik Hegadekatti [7] proposed "Democracy 3.0" which which stored votes on a public blockchain and was supposed to be transparent, anonymous, and verifiable. The author suggested a proof-of-stake consensus algorithm that allowed for low-cost transactions and high throughput, making it suited for use in large-scale elections.

"Agora", is a recent blockchain voting system [8]. This system employs a permissioned blockchain, which enables secure and transparent voting. The authors proposed a consensus technique based on a proof of work that is both secure and scalable. While these approaches have shown potential, deploying blockchain-based voting systems still presents hurdles.

Scalability is one of the most difficult issues. Blockchain-based solutions can be slow and costly, making them difficult to scale up for large-scale elections. Another difficulty is the issue of voter privacy. Chauhan et al. argues that while blockchain-based systems can provide openness and immutability, they can potentially jeopardize voter anonymity if not appropriately handled [9].

Regarding these difficulties, several academics, such as Bazzi et al. [10] and Hussain et al. [11], have proposed hybrid systems that combine the advantages of blockchain technology. For example, both recommended using blockchain technology for vote counting while employing traditional techniques for voter authentication and verification. For instance, Moubarak et al. [12] suggest that blockchain-based voting systems are vulnerable to attacks such as 51%. These can jeopardize the system's security. This happens when a single entity, or group of entities, controls more than half of the network's computing or hashing power. This means that they can potentially manipulate transactions.

Finally, blockchain-based voting systems have the potential to improve vote integrity and transparency. Yet, there are still issues to address, such as scalability and voter privacy.

Further research and development are required to produce a strong and reliable blockchain-based voting system suitable for large-scale elections.

*2.1. Market Study*

Voting systems have been an integral part of democratic societies for centuries. The emergence of blockchain technology has created new prospects for developing safe and transparent voting systems. Many blockchain-based voting systems have arisen in recent years, each with their own sets of advantages and drawbacks. This section will go through some of the most well-known blockchain-based voting systems on the market.

Horizon State [13] is a blockchain-based voting system that uses blockchain technology to enable safe, transparent, and efficient voting. The method enables voters to vote in elections using their cell phones or laptops, making it simple and accessible to all. Horizon State's underlying blockchain technology is Ethereum, which provides immutable and transparent evidence of all transactions. The system is planned to be simple to use, with a straightforward user interface that takes voters through the voting process.

Another blockchain-based voting system that aspires to improve the way we vote is Follow My Vote [14] . The system is supposed to be safe, transparent, and accessible to all users. Follow My Vote employs blockchain technology to verify that each vote is correctly saved and cannot be manipulated. Voters may vote from anywhere using their cell phones or laptops, making it simple and accessible for everyone. Follow My Vote is based on the BitShares blockchain and provides a quick and efficient voting mechanism.

Voatz [15] is a blockchain-based voting system that ensures voter identity through biometric authentication. The system is meant to be safe, transparent, and accessible to all users. Voatz employs blockchain technology and biometric verification to ensure that each vote is correctly saved and cannot be tampered with. Voters may vote from anywhere using their cell phones or laptops, making it simple and reachable for everyone. Voatz is a scalable and efficient voting platform built on the Hyperledger Fabric blockchain.

Blockchain-based voting systems have the potential to change the way we vote forever. These mechanisms are meant to be safe, transparent, and accessible to anyone. They use blockchain technology to verify that each vote is correctly saved and cannot be tampered with. While each of the initiatives mentioned above has its own set of advantages and disadvantages, they all represent an important step forward in the development of safe and transparent voting systems.

*2.2. Blockchain Concepts*

2.2.1. Blockchain

Blockchain technology is a distributed, decentralized digital ledger that allows for secure data storage and exchange. It is an innovative breakthrough that has grown in prominence since the appearance of Bitcoin, the first decentralized digital currency, and has the potential to disrupt numerous businesses.

> "A blockchain is essentially a distributed database of records or public ledger of all transactions or digital events that have been executed and shared among participating parties" by Puranik et al. [16].

A Blockchain is a network of interconnected blocks, each containing a cryptographic hash of the preceding block and a set of transactions. The cryptographic hash function is an essential part of blockchain technology because it ensures that any changes made to a block are discovered and rejected by the network. This makes the blockchain tamper-proof, as any modifications to one block require changes in all the following blocks.

The blockchain technology is decentralized, so no single entity controls the network, allowing for greater transparency and security. The network is instead maintained by a distributed network of nodes, each with a copy of the blockchain. Mining is a process that validates and adds transactions to the blockchain by solving challenging mathematical puzzles to verify the transaction and add it to the blockchain. In cryptocurrency, since it uses a blockchain that is decentralized and distributed, there is no need for intermediaries

such as banks or financial institutions to facilitate transactions. This has the potential to significantly cut transaction costs while increasing efficiency, especially in businesses such as banking.

### 2.2.2. Smart-Contracts

SSs, which are self-executing programs running on blockChain platforms, have gained significant attention due to their potential to automate contractual agreements [17,18]. However, SSs are also prone to errors and vulnerabilities that can result in significant financial losses. Therefore, formal methods have been proposed as a solution for verifying the correctness and security of SSs. By mathematically proving the correctness of SSs, formal methods can increase confidence in their performance and ensure that they function as intended. The use of formal methods for SS verification is an active research area with promising results, and it is expected to become a crucial component of blockchain technology.

### 2.2.3. Transactions

Transactions refer to the exchange of assets or information between parties, which can be recorded in a blockchain. As described in the article "Towards Anonymous, Unlinkable, and Confidential Transactions in Blockchain" [19], transactions in a blockchain can be designed to be anonymous, unlinkable, and confidential. This means that the identities of the parties involved in the transaction can be hidden; the transaction cannot be linked to other transactions, and the details of the transaction can be kept private. Transactions are a fundamental component of blockchain technology, allowing for secure and transparent exchanges without the need for intermediaries such as banks.

### *2.3. Comparing Blockchain Projects*

Table 1 describes the comparison between three existing blockchain-based voting systems: Horizon State [13], Follow My Vote [14], and Voatz [15]. These blockchain-based voting systems have similar aims of assuring safe and transparent voting procedures. Each system has distinguishing qualities that make it distinct from the others.

- Voter anonymity: This refers to the principle that a voter's identity should be kept secret and not be revealed to anyone, including the election officials. In other words, it means that no one should be able to link a particular vote to a specific voter.
- End-to-end verifiability: This is a property of e-voting systems that allows voters to verify that their votes have been correctly recorded and counted. It means that voters can check that their votes have not been altered or tampered with during the voting process and that they have been included in the final tally.
- Accessibility: This is the ease with which one person was able to access and use one system, regardless of their abilities. This demonstrates how easily the voters can vote and also understand what is happening in each state of the system.
- Security: This refers to the measures or procedures taken to protect something from threats or unauthorized access, that is, how hard it is for someone to violate the system. For example, it determines how hard it is for the voter to manipulate the election. The harder the work, the more secure the system is.
- Scalability: This means the ability of a system to handle increasing amounts of data or workload without compromising its performance. This means that the bigger the poll, the more scalable the system needs to be. This measure copes with the number of voters the system handles at the same time.
- Transparency: This represents the quality of the system state being easily visible or understood. This means the processes and transactions carried out within the system are easily traceable, and the information related to them is readily available to all authorized voters.
- Decentralization: This refers to the process of distributing power or control away from a central authority or entity, and instead distributing it among multiple nodes or

participants in a network. Thus, multiple nodes have to validate a transaction (voters vote) before being approved and registered.

- Hybrid consensus mechanism: This is a combination of two or more methods used to achieve consensus in a blockchain network. It is planned to mitigate some weaknesses of a specific consensus algorithm, by changing that part of the algorithm with another algorithm, to make it more secure and as efficient as possible. The consensus in a blockchain is the process of achieving agreement among all the nodes, for example, approving a transaction.
- Scheduler generation: This relates to the process of creating an optimal schedule, taking into account all the constraints created.
- Register customized constraints: This concerns the ability of customization and whether a user can customize the schedules and the configuration of the blockchain itself.

**Table 1.** Comparison of blockchain projects.

| Feature | Horizon State [13] | Follow My Vote [14] | Voatz [15] |
|---|---|---|---|
| Voter anonymity | X | X | X |
| End-to-end verifiability | X | X | X |
| Accessibility | X | | X |
| Security | X | X | X |
| Scalability | | | X |
| Transparency | X | X | X |
| Decentralization | | | X |
| Hybrid consensus mechanism | | | X |
| Scheduler generation | | | |
| Register customized constraints | | | |

In terms of voter anonymity, Follow My Vote [14] and Voatz [15] provide end-to-end encryption and blockchain anonymity, while Horizon State offers encrypted and private voting but not necessarily anonymity. End-to-end verifiability is a vital feature provided by all three systems, allowing voters to prove that their vote was correctly counted. Accessibility is also a goal for these systems, with Voatz having mobile voting capabilities and Horizon State and Follow My Vote providing accessible voting alternatives for those with impairments. The security of any voting system is critical, and all three blockchain-based systems use powerful encryption technology to protect against hackers and manipulation. Horizon State [13] and Voatz [15] have an advantage in terms of scalability, with Horizon State's [13] hybrid consensus method and Voatz's [15] proprietary mobile voting technology both enabling more efficient and quicker vote processing.

Transparency is a feature shared by all three of these systems, with each giving immutable recordings of the voting process. Another crucial consideration is decentralization, with Follow My Vote [14] and Voatz [15] employing a completely decentralized method, while Horizon State [13] employs a more centralized architecture with distributed components.

In terms of security, both sides employ encryption and digital signatures, but the proposed solution adds a layer of security through its hybrid consensus mechanism. This makes it less prone to single-point-of-failure attacks and increases its overall security.

In terms of cost, the state-of-the-art solutions are expensive due to their high energy consumption, while the proposed solution reduces costs by utilizing a more energy-efficient consensus mechanism.

Finally, accessibility is limited in the first system due to its technical complexity, while the proposed solution uses a user-friendly interface to increase accessibility. Further information on the proposed analyzed system may be consulted in Appendix A.

In the next section, we will provide further information on the proposed voting system. It mainly distinguishes it from existing blockchain initiatives by combining a genetic algorithm-based scheduling system with a voting system, creating a solution for choosing suboptimal scheduling possibilities within voting interventions.

## 3. Proposed System Architecture and Design

Figure 1 illustrates the architecture of the complete system, in which there are three parts represented:

1.  External API: This is the part that the user interacts with. This component is in charge of collecting the restrictions and the voting parameters, as well as showing the poll and their results through a web interface. It will be through this that the users will vote and interact with the blockchain.
2.  Algorithm: In this algorithm, all the heavy processes of generating the schedules occur. It processes all the constraints that come from the external API and uses a genetic algorithm to generate the schedules.
3.  Voting Blockchain: This module represents the blockchain that handles all the logic of our voting system. Each purple circle represents a node from the blockchain, and all the nodes are in their own isolated environment, preferably in different servers.

To achieve the goals of our proposed system, we have developed a hybrid architecture that combines the best of both centralized and decentralized systems. The architecture consists of two main parts: the schedule generation system and the voting system.

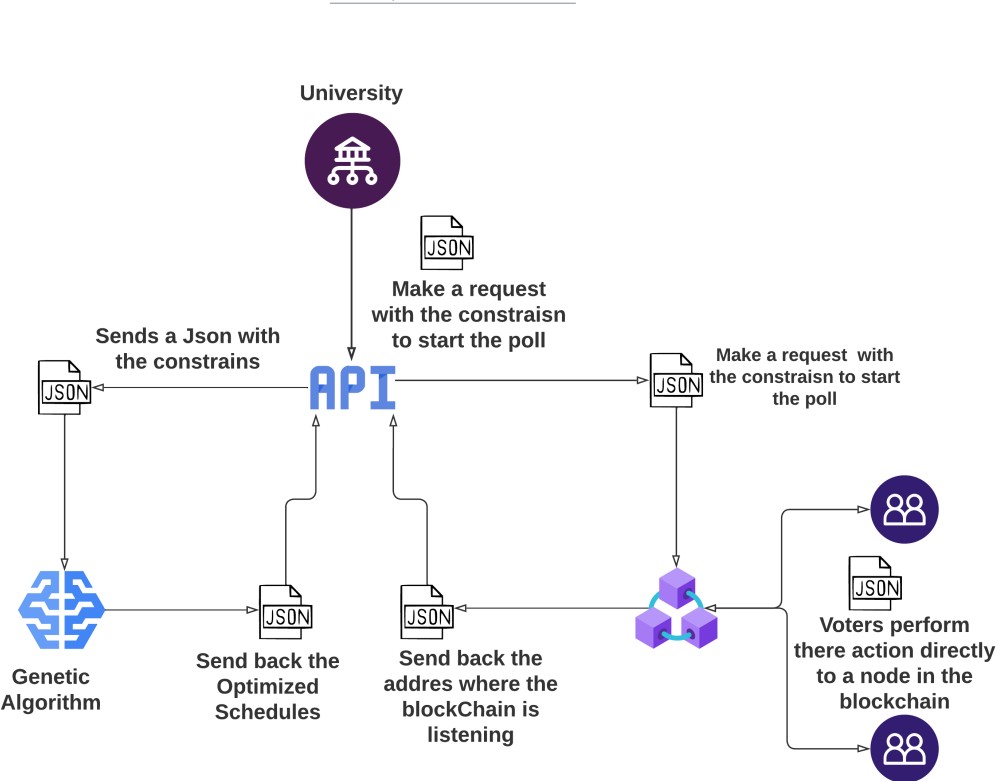

**Figure 1.** Proposed System Architecture.

The schedule generation system is based on a genetic algorithm, which is a powerful optimization technique inspired by the process of natural selection. The genetic algorithm

generates optimal schedules by iteratively improving candidate solutions through selection, crossover, and mutation operations. The system takes into account multiple constraints, such as teacher availability, student preferences, and course requirements, to generate schedules that satisfy all stakeholders.

The voting system is built on top of a blockchain network, ensuring transparency, immutability, and security of the voting process. The blockchain network is projected to be permissioned, meaning only authorized parties are allowed to participate in the network. The consensus mechanism used a hybrid approach between proof of work and proof of authority. This is a more efficient and scalable alternative to only proof of work. This approach consists of one node validating the transactions and generating a list of authorized nodes, and then these authorized nodes use their authority to finalize the block and reach a consensus of accepting the block or rejecting it.

To interconnect the voting system and the schedule generation system, we have developed a custom API that enables communication between the two systems. The API allows the schedule generation system to receive voting data from the blockchain network, process the data, and generate schedules based on the results. The schedules are then securely stored on the blockchain network, ensuring the immutability and transparency of the scheduling process.

Overall, our hybrid architecture offers a secure, efficient, and flexible solution for both voting and schedule generation. By combining the strengths of blockchain and genetic algorithms, we have created a system that can be customized to fit the needs of any educational institution.

*3.1. Scheduling System*

The scheduling system is projected to provide an efficient and flexible scheduling solution that can handle complex scheduling problems. It is built on a genetic algorithm, which is a powerful optimization technique that is inspired by the process of natural selection. The scheduling system consists of several components:

- Data Collection: The data collection component is responsible for collecting all the relevant data required for the scheduling process. This includes the course schedules, room availability, and instructor preferences.
- Fitness Function: The fitness function is the core component of the scheduling system. It evaluates the fitness of each schedule and sets a fitness score based on a group of criteria. This component is crucial for guiding the evolution of the schedules toward optimal solutions.
- Genetic Operators: The genetic operators are responsible for creating new solutions by applying genetic operators such as crossover, mutation, and selection. These operators mimic the process of natural selection and help to generate new and better schedules.
- Evolutionary Algorithm: The evolutionary algorithm is the overarching component that combines all the other components to create a complete scheduling solution. It is responsible for guiding the evolution of the schedules toward optimal solutions.

Figure 2 represents all the major steps that the scheduling system carries out. The first step is collecting and processing the constraints that are received from an external API. These constraints come in a JSON format and the system maps to classes that represent our entity, such as the teachers, the rooms, and so on. When the classes are mapped, the system starts by building our population, which means creating grids that represent our schedule, where for the first populations the classes are randomly placed in the grid respecting only the hard constraints, creating as many populations as specified in the configuration parameters. This value varies depending on the capacity of the server, and the more powerful the server, the more populations will be created.

Then, the fitness is calculated; this is a way of rating all the populations where the more soft constraints there are, the higher the score is. The next step is to save the schedules with the highest scores; the number of schedules saved is the number of results for the poll. Before it moves to the next step, our system evaluates the saved schedules if all the

programs have a score superior to the required score, which is the minimum score that a schedule needs to have to be able to be considered as a result of the pool. If there are enough schedules, we can finish the process and notify the API that we have found the results for the pool. If not, we continue to the next step.

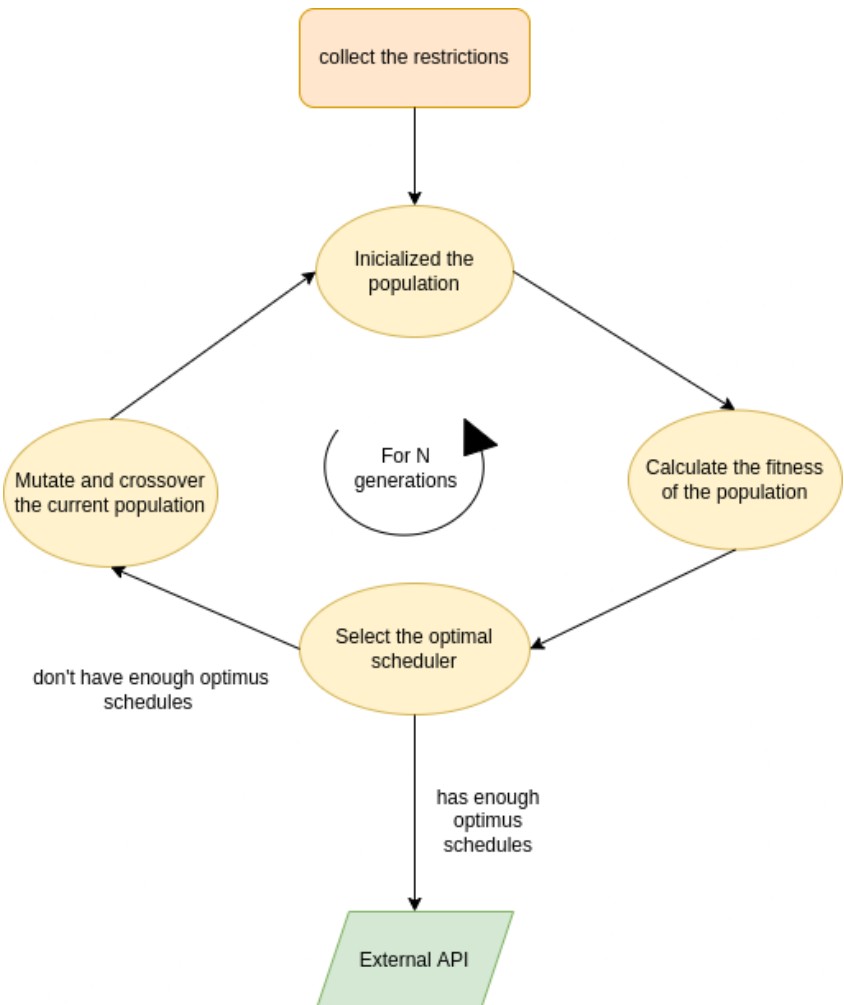

**Figure 2.** Genetic algorithm diagram.

The next step is the crossover and mutation step, which is when the schedules of the population are mixed together, the crossover, and randomized the mixed grid, the mutation, to create a new population, taking into consideration that all the newly generated schedules always need to respect the hard constraints. The process of a schedule that runs through all of these steps is called a generation. A schedule can run as many generations as specified in the configuration parameters of the genetic algorithm. The more capable the server, the higher the number of generations.

Creating a timetable for an educational institution may be a difficult and time-consuming task, especially when multiple considerations such as usable resources, course needs, and student preferences are taken into account. In our system, a genetic algorithm is given for timetable generation in order to automate this process and deliver an ideal answer.

A genetic algorithm is a search heuristic inspired by the natural selection process. It entails using a collection of genetic operators such as crossover and mutation to develop a population of solutions toward an ideal solution. The evolutionary algorithm has seen widespread use in various optimization issues, including scheduling challenges. Several studies have reported successful implementations of genetic algorithms in timetable generation, including [20,21]. These studies have shown that genetic algorithms can provide

efficient and effective solutions for timetable generation, especially when considering complex constraints and preferences.

The genetic Algorithm 1 is used in our system to build an ideal timetable that meets numerous limitations such as course timings, teacher availability, and classroom capacity.

---

**Algorithm 1** A genetic algorithm

---

**Require:** $100 \leq Generations$                 ▷ The maximum iterations that the algorithm can do
**Require:** $30 \leq Population$                 ▷ The number of optimal solutions per generation
**Require:** $MinimumScore \geq 80$         ▷ The minimum score to the schedule by approved
**Require:** $Nsolutions \geq 5$     ▷ The number of approved schedules to the program stopped
  $currentGeneration = 0$
  $appreovedSchedules = 0$
  **while** $currentGeneration < Generations || appreovedSchedules < Nsolutions$ **do**
    **function** INITIALIZATION(constrains)
    **end function**
    **function** FITNESS EVALUATION(generated schedules)
    **end function**
    **function** SELECTION(generated schedules, Minimum Score )
    **end function**
    $appreovedSchedules = getAprovedSchedules(generatedschedules)$
    **if** $appreovedSchedules \geq Nsolutions$ **then**
      $EndAlgorithm$
    **end if**
    **function** CROSSOVER(generated schedules)
    **end function**
    **function** MUTATION(generated schedules)
    **end function**
    **function** ELITISM(generated schedules)
    **end function**
  **end while**

---

The following steps are included in the schedule-generating process:

1. Initialization: A population of random timetables is generated.
2. Fitness evaluation: Each timetable in the population is evaluated based on its fitness function, which considers various constraints and preferences.
3. Selection: A set of parent timetables is selected based on its fitness.
4. Crossover: The selected parent timetables are combined to generate a new set of timetables.
5. Mutation: The new set of timetables is subjected to mutation, where certain genes (i.e., classes, instructors, or classrooms) are randomly changed.
6. Elitism: The best timetables from the previous generation are preserved in the current generation to ensure convergence toward the optimal solution.
7. Termination: The algorithm terminates when a stopping criterion is met, such as a maximum number of generations or a satisfactory fitness level.

The default settings that should be used vary depending on the system; however, several metrics should be considered. The mutation rate should be low enough to maintain the best solutions identified, but high enough to explore new areas of the search space. A mutation rate of 0.01 to 0.1 is commonly utilized. The population size should be big enough to preserve demographic variety without becoming computationally costly. A population size of 50 to 100 people is commonly utilized. The number of generations should be determined by the desired level of convergence and the available computer resources. The minimum fitness score is set to the default of 85, on a scale from 0 to 100, with 0 as the minimum and 100 as the maximum. This parameter should be changed depending on the

results. If the result is far from the expected score, it should be increased. However, if it is taking too long to generate a schedule, the score should be decreased.

The proposed genetic algorithm for timetable generation is expected to provide an efficient and effective solution for the timetable generation process in our system, and is expected to significantly reduce the time and effort required for manual timetable generation.

### 3.2. Voting System

The voting system is projected to provide secure, reliable, and transparent voting. It is built on a blockchain platform, which ensures that every vote is saved in a tamper-proof manner. The voting system consists of several components:

1.  Voter Registration: To participate in the voting process, users need to register and authenticate themselves using a secure and user-friendly interface. This component includes identity verification and digital signatures to ensure the authenticity of each voter.
2.  Block Creation: The block creation component is responsible for creating the block and ensuring that it contains all the necessary information required for the voting process. This includes candidate lists, referendums, and voting rules.
3.  Voting: The voting component is responsible for ensuring that each vote is recorded and stored in the blockchain securely and transparently. The component ensures that the vote is valid and that the voter has not voted twice.
4.  Vote Counting: The vote counting component is responsible for tallying the votes and declaring the winner. This component ensures that the vote count is accurate, transparent, and tamper-proof.

Figure 3 represents all the steps that a vote needs to pass before being inserted in blockchain. The suggested system is a blockchain-based voting platform that is safe and encrypted, allowing organizations to develop and configure their bespoke system for holding various polls. The major goal of this system is to provide a transparent and tamper-proof voting process that removes the potential of hacking while maintaining voter anonymity.

To perform this, the system makes use of a customized blockchain, which eliminates the requirement for institutions to pay fees to publish on blockchains such as Ethereum or Bitcoin. The system may be built as Docker [22] images that are deployed with Kubernetes [23] to produce numerous instances per server on as many servers as the institution requires, making it cost-effective. Kubernetes is critical to ensure that the system is always highly available and responsive. Nonetheless, the system's deployment component may be tailored to the institution's present system. The system validates itself via proof of work, which is more expensive on the computer but a safer alternative for tiny blockchains. There is no requirement for staking or the use of a coin or token with proof of work validation.

The online interface for establishing and editing polls allows the pool's developer to assign a height to each person's vote, which is essential in administration board polls where each person's vote has a varied amount of importance. Its height can be described as the proportion of the institution that the individual owns, such as in an administrative pool. To ensure transparency, the system offers anonymous voting, where no one knows who owns the wallets, but everyone can see where each wallet voted. Additionally, the blockchain is encrypted to guarantee that data are safe and that only authorized parties have access to data.

Overall, the suggested system is a highly scalable, secure, and cost-effective voting platform that can be tailored to institutions' unique demands. The system assures that the voting process is visible, tamper-proof, and anonymous by utilizing blockchain technology, making it appropriate for a variety of applications where trust and transparency are crucial.

The construction of an SS on a blockchain containing a list of voters and their votes, all encrypted using homomorphic encryption (HE). It is a type of encryption that allows calculations to be conducted on the ciphertext. This leads to an encrypted result that may

be decoded to obtain the same result [24]. Each voter would be given a one-of-a-kind private key that would be used to encrypt their vote.

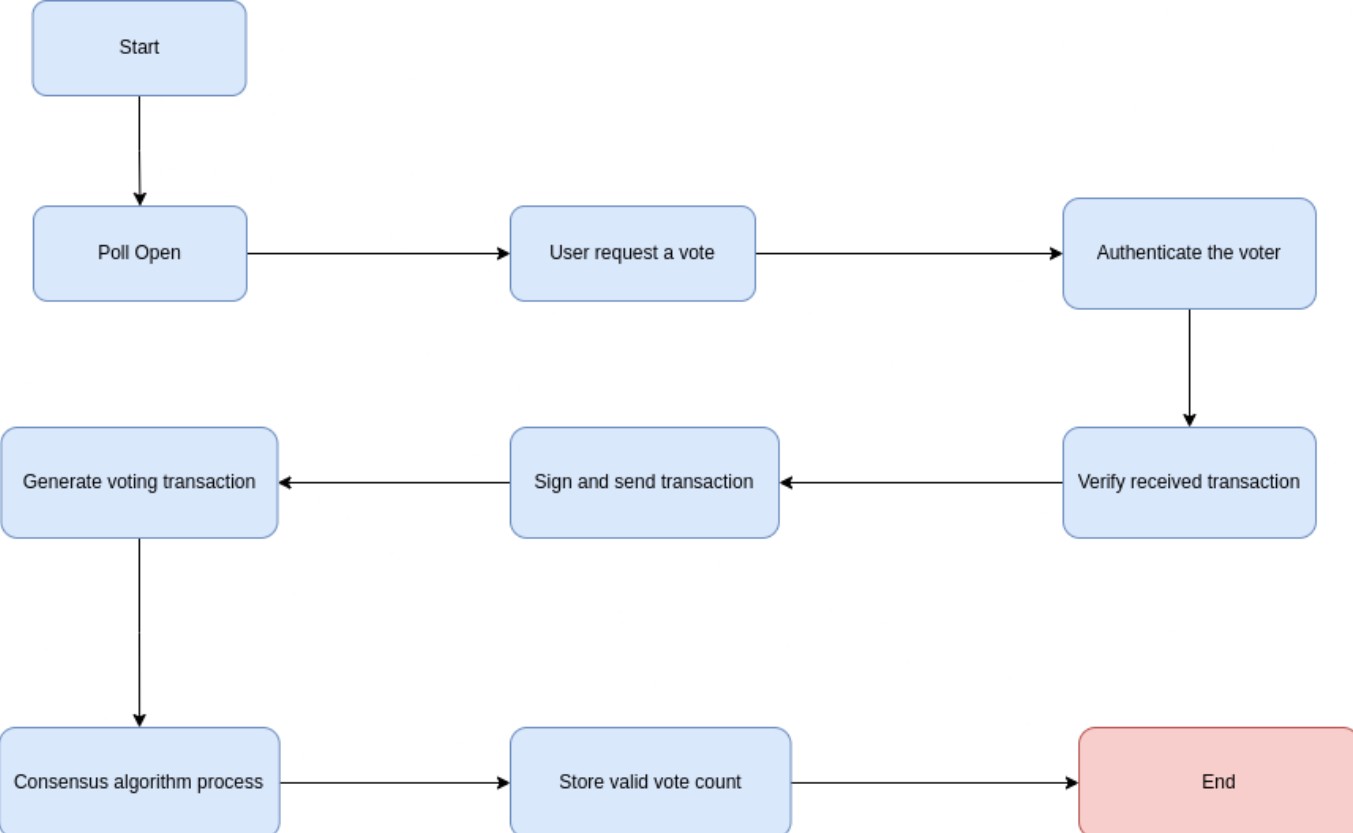

**Figure 3.** Blockchain operation flowchart.

The encrypted votes would be transferred to a mixer, which would use the mixing process to shuffle the votes. Depending on the application, the mixer might be centralized or decentralized. The use of mixing increases privacy and helps to avoid vote rigging.

Figure 4 represents all the steps that the mixer undergoes, which ensures that the votes stored in the blockchain are not in the same order as they were voted. This flowchart is for a centralized mixer. In a decentralized mixer, the steps are the same but have a few extra steps that divide the votes into an equal number of votes for each node. Then, in the end, it has a merged process to ensure that all nodes have the same block order. A decentralized mixer is only recommended when many voters exist.

The SS would be set up to employ zero-knowledge proofs (ZKPs) to allow each voter to verify their vote without exposing their decision. A ZKP is a cryptographic procedure that enables one party to demonstrate knowledge of a secret without disclosing the secret itself [25]. The Schnorr protocol, for example, is a ZKP that may be used to establish knowledge of a discrete logarithm without exposing it [26].

The SS would also be set up to employ ring signatures, which are made by several voters but only one of them is the real signer. This makes determining which voter signed the message challenging, as all members of the group might have been responsible for the signature. Rivest et al. [27] pioneered the use of ring signatures in different privacy-preserving protocols, including anonymous electronic payment systems. Ring signatures are a form of digital signature system that allows a group of users to sign a message without exposing which member of the group signed it [28].

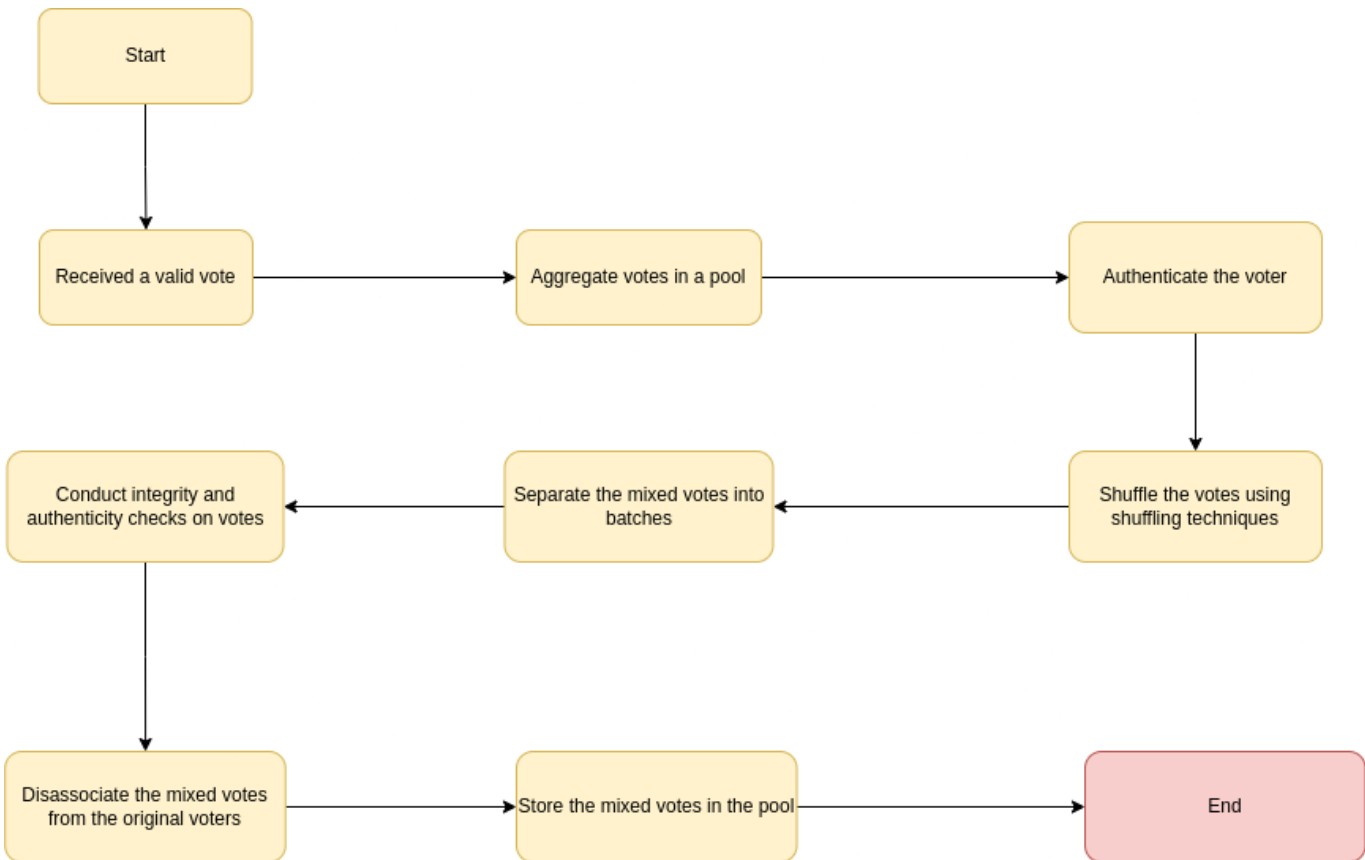

**Figure 4.** Mixer Flowchart.

Upon the completion of the voting, the SS with the list of encrypted votes will be used to count the votes using HE. This permits the votes to be tabulated without exposing each voter's preference. This technique allows each voter to know who they voted for but not where their fellow voters voted, preserving anonymity. It also enables votes to be tabulated without disclosing each voter's decision, respecting their privacy and security. The use of zero-knowledge proofs and ring signatures adds a degree of security and anonymity, making it even more difficult for third parties to learn the identities of voters or alter votes.

*3.3. Methodology*

The methodology adopted in this study was divided into three parts: (1) producing optimum schedules using a genetic algorithm, (2) constructing a decentralized voting system using a local blockchain, and (3) offering a user interface for adjusting the algorithm and voting system parameters.

The study utilized real-world data from a university, such as a list of instructors, class schedules, and classroom features from the previous year. To analyze the adaptability of the algorithm, a set of restrictions was constructed and the algorithm's capacity to comply with the requirements was evaluated.

In the second part of the study, a local blockchain was initialized, and several users selected their preferred schedules. The blockchain was tested for its performance and integrity by attempting to manipulate it, such as by multiple voting, trying to pass for another user, and submitting numerous votes at once.

The results were satisfactory, indicating that the generated schedules could be implemented in such a scenario, and the blockchain's integrity was not compromised. The study used a mixed-methods strategy that included genetic algorithms, blockchain technology, and user voting to choose optimum schedules and evaluate blockchain integrity. Real-world data were utilized to evaluate the system's applicability in a university context, making it a vital addition to the fields of educational scheduling and blockchain technology.

## 4. Results and Discussion

### 4.1. Scenario

In the context of a university that needs to generate schedules for all courses at the beginning of the semester, various restrictions need to be analyzed and considered. A problem with the traditional approach is that when this scheduler has been created, it is really difficult to consider every professor's restriction, and even worse is when those restrictions collide with others' limitations (e.g., room capacity, lecture overlap). Our system proposes a genetic algorithm to generate possible schedule solutions. We will try every smart combination of the classes, respecting the restriction, until we find a schedule that fits perfectly with the restriction. However, that is not always the case because of overlap restrictions; it is sometimes impossible to respect all the restrictions, more specifically the soft restrictions. Here we introduce the second part of our system, the blockchain voting system, which will select a specified amount of schedules to send to vote. The professors may choose, through a poll, the schedule that they prefer. All those votes will be stored in a blockchain to secure all the fairness and transparency that a poll should have.

Back to the beginning, the first step is to register all the restrictions that the university has. These restrictions may include the number and capacity of classrooms, the specifications of each classroom (e.g., the presence of projectors or computers), the availability of professors, and other relevant factors. Once these requirements have been identified and registered in the system, a JSON file will be created with this format.

The JSON is formatted as follows: first, the name of the constraint; second, the values the constraint can have. For example, the hour constraints, where their values mean the hours during which it is possible to have classes. The meaning changes from constraint to constraint and from context to context. A more simple constraint is represented as an array with values; we can have more complex constraints such as a constraint that depends on another constraint. For example, the class entity, for one class to happen, requires the teacher that is going to teach that class to be available at the time of the class itself. This is represented in the JSON by having a tag that directly references other constraints by their unique identifier. In this example, the unique identifier is the name of the teacher, and the tag class has a tag that references the name of the teacher (Figure 5).

We can also prioritize one entity face to another. That means that we can prefer having an entity before it happens to another entity. In this case, we can prefer having a theoretical class before having a practice class. In the JSON, this constraint is just a list that respects the order in which we want our classes to happen.

The scheduling system uses the specified requirements and parameters for the genetic algorithm, which can be adapted manually or automatically. These parameters include the maximum population size, maximum number of iterations, minimum score threshold, and pool size (i.e., the number of schedules to be generated). Upon completion of this process, a set of schedules is produced and presented in a specific format.

The last type of constraint is an entity that indirectly references another entity by having the same tag with the same value. For example, this is useful in the case where the professor needs a projector to teach a specific class. With this type of constraint, we only allow that class to exist in a classroom with a projector (Figure 6).

For this example, a fairly simple restriction problem is used in order to be more easily understood. The schedule generated is the table "Example of a schedule generated by the system" Table 2. The JSON file for this example is provided at the end of the article.

```json
{
    "Teacher": {
        "name": "Prof 1",
        "subject": "AED1",
        "hour_per_day": 10
    },
    "Schedule": {
        "Subject": "AED1",
        "Semester": "1",
        "Type": "Lab",
        "Teacher": "Prof 1",
        "Duration": "2",
        "Capacity": 15,
        "Projector": false,
        "Sockets": true
    }
}
```

**Figure 5.** Teacher, Schedule Constraints.

```json
{
    "Classrooms": {
        "Name": "Room 101",
        "Projector": true,
        "Sockets": true,
        "Capacity": 25
    },
    "Schedule": {
        "Subject": "AED1",
        "Semester": "1",
        "Type": "Lab",
        "Teacher": "Prof 1",
        "Duration": "2",
        "Capacity": 15,
        "Projector": true,
        "Sockets": true
    }
}
```

**Figure 6.** Classrooms, Schedule Constraints.

**Table 2.** Example of a schedule generated by the system.

| Timetable | Room 101 | Room 102 |
|---|---|---|
| Monday 9 h | None | AI Teo 2 |
| Monday 10 h | None | AI Teo 2 |
| Monday 11 h | AED1 Teo 1 | None |
| Monday 12 h | AI Prat 2 | AED1 Lab1 |
| Monday 13 h | None | AED1 Lab1 |
| Monday 14 h | None | None |

Following the schedule generation, a voting system is initiated, which can be either anonymous or not and can have equal or weighted voting, depending on the administrator's configuration.

Each voter is given access to a unique wallet through a public address (see Figure 7).

```
1DbKWGnAzf×8VxVwpCSoUGKXEQwo9qZPtL
1Hpi3f6JsmyKUapvcV62NKZkm8P5NFYET
1M8Fb6zbiBrmAePCb5fsTUZSFCRD2PTAUK
1Pj1KTdyqNQv1hphPWbXouZHh7TPTRJop
```

**Figure 7.** List of the public addresses of the wallet of the voters.

With this wallet, the voter has access to the blockchain and to the poll with the possibility to vote. When the voter votes and passes through all the authentication processes, a new transaction is created. The voter can see all the transactions made in the poll. For this system and for every transaction, a block is created in order to be more easily explainable and understandable, and in another example with many voters that is an easy optimization to implement. In Figure 8, we can see all the transactions made for this example with four voters and an extra transaction which is the genesis transaction, i.e., the transaction that initializes the pool.

```
Previous Hash: 00000157a2b5c04a3f4eed716d5c8eba68ee345a007b0c2343441112315767bb
Hash: 00013412232d1018be11689cbfbc7dbcddd8ed2191278dc36464fa26eaeb9e01
Pow: true
Previous Hash: 00009c94a5df539809bbfa706ee9c7a8f40f1f094c43745290f2748feaa7a027
Hash: 00000157a2b5c04a3f4eed716d5c8eba68ee345a007b0c2343441112315767bb
Pow: true
Previous Hash: 0000376aa3195fe59ced8f93250fb036158cb1ec37afd95ec96755413db25bd
Hash: 00009c94a5df539809bbfa706ee9c7a840f1f094c43745290f2748feaa7a027
Pow: true
Previous Hash: 00014c3e43bdb7bd0bb26112861a31ce74e6a1badfbc8cc3006a73e6c097e5de
Hash: 0000376aa3195fe59ced8f93250fb036158cb1ec37afd95ec96755413db25bd
Pow: true
Previous Hash: 000298a5a4100d5e706687f90a84575268b6d05d58cef0732a4fa43ef15c4868
Hash: 00014c3e43bdb7bd0bb26112861a31ce74e6a1badfbc8cc3006a73e6c097e5de
Pow: true
```

**Figure 8.** Blockchain Transactions.

In the end or during the poll, the voters with authorization and, depending on the poll configuration, the voter can see the result of the pool (see Figure 9).

```
Total votes of "Option 1": 1
Total votes of "Option 2": 3
```

**Figure 9.** Result of the poll.

*4.2. Discussion*

One of the biggest challenges of a system like this is that the findings of the unique anonymous voting mechanism detailed in this research are encouraging, but they should be considered with caution owing to the testing's hypothetical character. The suggested approach has several advantages and drawbacks when compared to the previously discussed blockchain-based voting systems.

One of the solution's merits is the use of HE to keep votes private while still allowing vote counting. This is a big advancement over certain previous blockchain-based voting systems, which do not provide complete secrecy. Furthermore, the proposed system's use of zero-knowledge proofs and ring signatures provides another degree of security and anonymity, making it more difficult for third parties to uncover the identities of the voters or influence the votes.

The proposed approach, however, has significant drawbacks, such as the requirement for a mixer to shuffle the encrypted votes. While this is a typical strategy in anonymous voting systems, it creates a possible point of failure and attack by relying on a centralized or decentralized mixer.

Additionally, due to a lack of resources, the testing of this method was merely empirical. While simulation findings seem encouraging, more testing and real-world implementation are required to assess the usefulness and efficiency of this system in a current voting environment. Moreover, using HE in the vote-counting procedure may impose some processing costs, potentially slowing down the voting process overall.

Furthermore, as compared to current blockchain-based voting systems, the suggested solution's combination of HE, zero-knowledge proofs, and ring signatures offers a solid basis for a safe and secret voting procedure.

## 5. Conclusions

In this study, we proposed a novel method that provides a full answer to the problems that organizations experience when creating schedules and conducting transparent and safe voting. The technology combines the efficiency and transparency of blockchain-based voting with the capability of genetic algorithms to design ideal timetables.

The system offers various advantages, including the potential to automate the scheduling process, lowering the stress on human schedulers. Furthermore, the implementation of blockchain-based voting protects voting process integrity, avoiding fraud and manipulation. The approach would be especially beneficial in organizations that demand a high level of security and transparency in their scheduling and voting procedures, such as government agencies, universities, and major enterprises. Because the system is integrated with existing scheduling software, it is also a viable option for firms wishing to streamline their scheduling operations.

The combination of the evolutionary algorithm and the voting system, in particular, makes it a perfect option for enterprises with a big and varied workforce, where establishing an optimal timetable might be difficult, and fair and transparent voting is crucial.

Overall, the suggested approach provides a one-of-a-kind answer to the issues that organizations are confronted with regarding scheduling and conducting safe and transparent voting. The use of modern technology such as genetic algorithms and blockchain-based voting makes it an excellent choice for enterprises trying to enhance their scheduling and voting processes while maintaining transparency and security.

*5.1. Key Contributions*

The authors propose a blockchain-based e-voting system that ensures security, transparency, and privacy. This approach may have the potential to progress the field of optimization

algorithms by applying genetic algorithms to the scheduling problem, which is a complicated and significant topic in many disciplines. They also present a detailed analysis of the proposed system's architecture and its components. The suggested approach provides a one-of-a-kind answer to the issues that organizations are confronted with regarding scheduling and conducting safe and transparent voting. The use of modern technology such as genetic algorithms and blockchain-based voting makes it an excellent choice for enterprises trying to enhance their scheduling and voting processes while maintaining transparency, security, and setups. An evaluation is provided to access the performance of the proposed system in terms of security and efficiency. A comparison is provided regarding other existing e-voting systems and highlights their advantages. Finally, they discuss the limitations of their proposed system and suggest future research directions to overcome these limitations. Overall, this paper provides solutions for secure and transparent e-voting using blockchain technology.

### 5.2. Limitations and Future Work

Notwithstanding the suggested system's achievements, some of its drawbacks must be acknowledged. For starters, the system depends on a genetic algorithm to generate optimal schedules, which may not be the only method/approach in some specific cases. To improve the quality of created schedules, alternative optimization procedures such as simulated annealing or tabu search might be considered.

Second, while blockchain technology provides transparency and immutability, it may have scalability challenges in large-scale settings. Further study is needed to investigate alternative methods to increase the voting system's efficiency in terms of scalability vs. computer consumption.

Third, the approach presupposes that all voters are truthful and that no collaboration exists among them. In a real-world setting, however, it is conceivable for a group of voters to conspire and affect the voting outcome. As a result, future research might look into how to prevent such conduct.

Finally, the existing approach does not take into consideration external variables that might disrupt the scheduling process, such as unplanned occurrences or changes in resource availability. Further development might involve including real-time monitoring and adaptation elements in the system to respond to changes in the environment and to ensure the resilience and robustness of the schedule creation and voting process.

**Author Contributions:** Conceptualization, B.M.B.P., C.P.d.A.S. and I.P.; methodology, B.M.B.P.; software, B.M.B.P.; validation, C.P.d.A.S. and I.P.; investigation, B.M.B.P., C.P.d.A.S. and I.P.; resources, B.M.B.P., C.P.d.A.S. and I.P.; data curation, B.M.B.P.; writing—original draft preparation, B.M.B.P.; writing—review and editing, B.M.B.P., C.P.d.A.S. and I.P.; supervision, I.P. and C.P.d.A.S.; reviewing, J.M.T., P.M.S. and R.S.M. All authors have read and agreed to the published version of the manuscript.

**Funding:** This work was partially supported by Base Funding—UIDB/00027/2020 of the Artificial Intelligence and Computer Science Laboratory—LIACC—funded by national funds through the FCT/MCTES (PIDDAC).

**Institutional Review Board Statement:** Not applicable.

**Informed Consent Statement:** Not applicable.

**Data Availability Statement:** Not applicable.

**Conflicts of Interest:** The authors declare no conflict of interest.

## Abbreviations

The following abbreviations are used in this manuscript:

| | |
|---|---|
| SS | Smart-contracts |
| e-voting | Electronic voting |
| He | Homomorphic encryption |
| ZKP | Zero-Knowledge Proof |
| CD-GA | Constraint-Directed Genetic Algorithm |
| Pow | Proof of Work |
| PoA | Proof of Authority |

## Appendix A

Table A1 describes how the proposed system differs and innovates relative to the systems already existing in the market in the most critical aspects. The state-of-the-art solutions employ encryption, multifactor authentication, and digital signatures to ensure security, while also providing a public ledger for tracking and verifying votes using blockchain technology. However, those systems had limited scalability due to high energy consumption and slow transaction processing times, leading to high costs and limited accessibility.

**Table A1.** Comparing the State-of-the-Art Solutions with the Proposed Solution.

| Criteria | State-of-the-Art Solutions | Proposed Solution |
|---|---|---|
| Security | Use of encryption, multifactor authentication, and digital signatures to ensure security | Hybrid consensus mechanism combining PoW [1] and PoA [2] for added security |
| Transparency | Provides a public ledger for tracking and verifying votes Uses blockchain technology | Uses blockchain technology for transparent and auditable voting |
| Scalability | Limited scalability due to high energy consumption and slow transaction processing times | Uses sharding to increase scalability |
| Cost | High energy consumption leads to high costs | Hybrid consensus mechanism reduces energy consumption and associated costs |
| Accessibility | Limited accessibility due to technical | Uses a user-friendly interface for increased accessibility |
| Reliability | Prone to single-point-of-failure attacks | Uses a decentralized network for increased reliability |

[1] PoW—Proof of work is a consensus technique used in blockchain networks in which members compete to solve challenging mathematical problems in order to validate and execute transactions, demonstrating their work and collecting incentives in exchange. [2] PoA—Proof of Authority is a consensus process in which network users are recognized and approved as validators to build and validate new blocks based on their reputation and identity rather than solving complicated cryptographic riddles as in Proof of Work.

The proposed solution, on the other hand, utilizes a hybrid consensus mechanism that combines Proof of Work (PoW) and Proof of Authority (PoA) for added security. It also uses blockchain technology for transparent and auditable voting, sharding to increase scalability, and a user-friendly interface for increased accessibility. Additionally, the hybrid consensus mechanism reduces energy consumption and associated costs, while using a decentralized network increases reliability. The proposed system also allows much more configuration and customization than the state-of-the-art solutions, where the voter can choose if they want anonymous voting or not, and in the same blockchain they can have different polls for different members of the public, which means that in some polls it is possible for everyone to vote and in other polls only an exclusive amount of people can

vote. Therefore, this can manage how big their system is, and they can increase their system just by deploying another docker image.

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
