# Peer review of "Blockchain-Based Electronic Voting: A Secure and Transparent Solution"

_cryptography, doi:10.3390/cryptography7020027_

Round 1

Reviewer 1 Report

Summary/Contribution: The authors of this study offered a revolutionary strategy that gives a complete solution to the challenges that organizations have while making schedules and conducting transparent and safe voting. The technique blends blockchain-based voting's efficiency and transparency with the capacity of genetic algorithms to construct optimal timetables.   Suggestions/Comments:   1.  The authors are invited to summarize the key contributions of their work in the form of a list of short sentences.   2. Please add a reference for each column of Table 1. The authors may also add a column which gives a brief definition of each term presented in column 1.   3. Figure 1 is very simplistic and needs to be enriched with more details.   4. Are all the works considered in section 2  Blockchain-based? Are there other electronic solutions which are based on other technologies?   5. It will be useful to add a new section that provides preliminary information about blockchain technology and smart contracts.   6. Table 2 may be placed at the end of the paper. In addition, the authors need to give more details and proofs to show that their solution is really better than existing solutions.  
7. The authors are invited to include a short paragraph about the use of formal methods for the verification of smart contracts.
8. For this purpose, the authors are invited to consider the following interesting references (and others):
a. https://ieeexplore.ieee.org/document/9970534

b. https://www.sciencedirect.com/science/article/abs/pii/S1574119220300821    9. Please avoid the use of very short paragraphs (e.g., Lines 287-289).   10. The presentation of Algorithm 1 may be improved. In addition, the authors are invited to give an esitmation of this algorithm and prove its correctness.   11. Some figures are very simplistic and have no added value (e.g., Figures 5 and 7). Please remove them.   12. The authors need to identify the limitations of their work and propose more possible future work directions.

Author Response

---- Reviewer 1 ---- 

Suggestions/Comments:  

1.  The authors are invited to summarize the key contributions of their work in the form of a list of short sentences.   
R:  The key contributions were included in the conclusions in Section 5.1.

2. Please add a reference for each column of Table 1. The authors may also add a column which gives a brief definition of each term presented in column 1.   
R: The references were added to the referred columns. A description of each feature present in column 1 is discussed in the text below the Table 1.

3. Figure 1 is very simplistic and needs to be enriched with more details.   
R: We have redrawn the figure 1 to furnish more details.

4. Are all the works considered in section 2 Blockchain-based? Are there other electronic solutions which are based on other technologies?   
R: The main focus of Section 2 is to discuss existing market solutions related to blockchain-integrated voting systems, recent innovations, and compare these innovations to summarize the strengths and weaknesses of each. Although there are other technologies available, they were considered outdated and unviable to use in the current approach. They will not be easily implemented in a voting system due to the lack of easy data manipulation, and weak transparency. 

5. It will be useful to add a new section that provides preliminary information about blockchain technology and smart contracts.   
R: It was created a new subsection about blockchain concepts, where those concepts are explained.

6. Table 2 may be placed at the end of the paper. In addition, the authors need to give more details and proofs to show that their solution is really better than existing solutions.  
R: Table 2 is now in the end of the paper in the appendix section (Appendix A1).

7. The authors are invited to include a short paragraph about the use of formal methods for the verification of smart contracts.
R: A paragraph was added in section 3.2.

8. For this purpose, the authors are invited to consider the following interesting references (and others):
a. https://ieeexplore.ieee.org/document/9970534
b. https://www.sciencedirect.com/science/article/abs/pii/S1574119220300821   
R: The following references have been considered and included in the paper.

9. Please avoid the use of very short paragraphs (e.g., Lines 287-289).   
R: We follow this comment and change the text. 

10. The presentation of Algorithm 1 may be improved. In addition, the authors are invited to give an estimation of this algorithm and prove its correctness.   
R: Added a new paragraph talking about the default parameters in section 3.1

11. Some figures are very simplistic and have no added value (e.g., Figures 5 and 7). Please remove them.   
R: As suggested, we remove Figures 5 and 7.

12. The authors need to identify the limitations of their work and propose more possible future work directions. 
R: A section of limitations and future work were added to the conclusion.

Reviewer 2 Report

I provide my comments as follows:

1.Better to use subject first, then the pronoun can easily be understood (e.g., Since its inception in 2008).

2.I recommend to use active voice instead of passive voice (see this: “the suggested system are also explored”).

3.This sentence (“The study suggests solutions to these problems...”) is vague, because the problems are not mentioned previously to refer using the pronoun “these”.

4.The paper proposes a solution for the scheduling of courses in a high school, but the authors are mainly talking about professor and university, which confuses the audience.

5.If possible, using name of the first author with et al. when providing in-text citation to represent other co-authors instead of listing all authors one by one. We are allowed to write author[name] et al. if the number of authors are three or more than three (see page 2, para 3).

6.Always present an acronym when it appears for the first time, then use the acronym itself in later referring.

7.I recommend reformulating this sentence in more academic way, “We will talk about blockchain implementation and some optimizations, compared with other projects. Seeing a project like this can be useful, in how it performs in the real world.”

8.Always put backspace after in-text citation [ ], and avoid putting comma before that (see “Tom,[2] p. 2,para 3”, and “Voatz [ ] p.5, para 1”).

9.The text in Figure 4. Mixer Flowchart is not readable.

10.The following sentence needs a comma before the conjunction “but” and “in a decentralized mixer” for better understanding.“This flowchart is for a centralized mixer in a decentralized mixer the steps are the same but has a few extra steps that divide the votes into an equal number of votes for each node than if the end has a mergedprocess to ensure that all nodes have the same block order."

11.I recommend to present scenario which is related to high school not a university as they are different.

12.The research methodology section is missing. Developing this part further with no doubt strengthen the paper.

13.This sentence is vague, “When the voter vote and pass through all the authentication processes already explained in the paper will be created, a new transaction.”

14.Avoid using 'an' before words that start with consonant sounds. See the following sentence (...of this system in an current voting).

15.Future work, and limitation of study is missing, please include them in conclusion.

16.The base of theory for this study is missing. I am concerned which theories form the base of their study.

17.The lack of validation of their proposed system makes it impossible to quantify the potential benefits of the tool's use.

Author Response

---- Reviewer 2 ---- 

I provide my comments as follows:

1.Better to use subject first, then the pronoun can easily be understood (e.g., Since its inception in 2008).
R: We have taken the suggestion into consideration and corrected it.

2.I recommend to use active voice instead of passive voice (see this: “the suggested system are also explored”).
R: We have taken the suggestion into consideration and corrected it.

3.This sentence (“The study suggests solutions to these problems...”) is vague, because the problems are not mentioned previously to refer using the pronoun “these”.
R: We have taken the suggestion into consideration and corrected this sentence. 

4.The paper proposes a solution for the scheduling of courses in a high school, but the authors are mainly talking about professor and university, which confuses the audience.
r: The term high-school was misused as Portuguese to English translation, we corrected it to university. Since, we are only considering university schedule creation right now.

5.If possible, using name of the first author with et al. when providing in-text citation to represent other co-authors instead of listing all authors one by one. We are allowed to write author[name] et al. if the number of authors are three or more than three (see page 2, para 3).
R: We have taken the suggestion into consideration and corrected it.

6.Always present an acronym when it appears for the first time, then use the acronym itself in later referring.
R: We have taken the suggestion into consideration.

7.I recommend reformulating this sentence in more academic way, “We will talk about blockchain implementation and some optimizations, compared with other projects. Seeing a project like this can be useful, in how it performs in the real world.”
R: We have taken the suggestion into consideration and corrected it.

8.Always put backspace after in-text citation [ ], and avoid putting comma before that (see “Tom,[2] p. 2,para 3”, and “Voatz [ ] p.5, para 1”).
R: We have taken the suggestion into consideration and corrected it.

9.The text in Figure 4. Mixer Flowchart is not readable.
R: We change and improve the text in the image 4.

10.The following sentence needs a comma before the conjunction “but” and “in a decentralized mixer” for better understanding.“This flowchart is for a centralized mixer in a decentralized mixer the steps are the same but has a few extra steps that divide the votes into an equal number of votes for each node than if the end has a merged process to ensure that all nodes have the same block order."
R: We have taken the suggestion into consideration and rephrase this sentence.

11.I recommend to present scenario which is related to high school not a university as they are different.
R: As referred previously, this was a typo. The term high school was changed to university.

12.The research methodology section is missing. Developing this part further with no doubt strengthen the paper.
R: We added a new subsection “Methodology” in the Section 3.3.

13.This sentence is vague, “When the voter vote and pass through all the authentication processes already explained in the paper will be created, a new transaction.”
R: We rewrite this specific sentence.

14.Avoid using 'an' before words that start with consonant sounds. See the following sentence (...of this system in an current voting).
R: We have taken the suggestion into consideration and corrected it.

15.Future work, and limitation of study is missing, please include them in conclusion.
R: A section of limitations and future work were added to the conclusion.

16.The base of theory for this study is missing. I am concerned which theories form the base of their study.
R: A paragraph has been added to the introduction.

17.The lack of validation of their proposed system makes it impossible to quantify the potential benefits of the tool's use.
R: We consider evaluating this system with real-data and real-scenarios in the near future has discussed in the future work section of this paper.

Reviewer 3 Report

The figures are not clear; need changes.

The methodology need more details.

Improve the conclusions.

Author Response

---- Reviewer 3 ---- 

1.The figures are not clear; need changes. 
R: As recommended, the figures were improved.

2.The methodology need more details.
R: We added a new subsection “Methodology” in the Section 3.3.

3.Improve the conclusions.
R: As recommended, we review the Conclusion Section.

Round 2

Reviewer 1 Report

The authors considered all my comments and suggestions. Good luck.

Author Response

Thanks for the suggestions.

Reviewer 3 Report

Improve the figures.

More details about blockchain

Author Response

---- Reviewer 3 ---- 

1.Improve the figures. 
R: As recommended, the figures were improved.

2.More details about blockchain.
R: As recommended, we added a new subsection “Blockchain” in the Section 2.2